# Exploring the Potential of Fecal Microbiota Transplantation as a Therapy in Tuberculosis and Inflammatory Bowel Disease

**DOI:** 10.3390/pathogens12091149

**Published:** 2023-09-09

**Authors:** Adrian Boicean, Dan Bratu, Sorin Radu Fleaca, Gligor Vasile, Leeb Shelly, Sabrina Birsan, Ciprian Bacila, Adrian Hasegan

**Affiliations:** 1Faculty of Medicine, Lucian Blaga University of Sibiu, 550169 Sibiu, Romania; adrian.boicean@ulbsibiu.ro (A.B.); radu.fleaca@ulbsibiu.ro (S.R.F.); sabrina.marinca@yahoo.com (S.B.); ciprian.bacila@ulbsibiu.ro (C.B.); adrian.hasegan@ulbsibiu.ro (A.H.); 2County Clinical Emergency Hospital of Sibiu, 550245 Sibiu, Romania; vasi_gligor@yahoo.com (G.V.); shellyleeb@yahoo.com (L.S.)

**Keywords:** fecal microbiota transplantation, FMT, tuberculosis, TB, inflammatory bowel disease, IBD, gut microbiota, immunomodulation, dysbiosis, adjunct therapy

## Abstract

This review explores the potential benefits of fecal microbiota transplantation (FMT) as an adjunct treatment in tuberculosis (TB), drawing parallels from its efficacy in inflammatory bowel disease (IBD). FMT has shown promise in restoring the gut microbial balance and modulating immune responses in IBD patients. Considering the similarities in immunomodulation and dysbiosis between IBD and TB, this review hypothesizes that FMT may offer therapeutic benefits as an adjunct therapy in TB. Methods: We conducted a systematic review of the existing literature on FMT in IBD and TB, highlighting the mechanisms and potential implications of FMT in the therapeutic management of both conditions. The findings contribute to understanding FMT’s potential role in TB treatment and underscore the necessity for future research in this direction to fully leverage its clinical applications. Conclusion: The integration of FMT into the comprehensive management of TB could potentially enhance treatment outcomes, reduce drug resistance, and mitigate the side effects of conventional therapies. Future research endeavors should focus on well-designed clinical trials to develop guidelines concerning the safety and short- and long-term benefits of FMT in TB patients, as well as to assess potential risks.

## 1. Introduction

Tuberculosis (TB) is defined as an infectious disease primarily produced by *Mycobacterium tuberculosis (Mtb)*, with a predilection for the respiratory system, particularly the lungs, but it can also infect other organs, leading to extrapulmonary tuberculosis. Although, nowadays, in developed countries, the TB incidence is reduced significantly thanks to various programs that prevent population infection, first of all through vaccination and numerous other national initiatives to prevent its spread, its prevalence still remains high, particularly in developing countries. Globally, TB ranks as one of the leading causes of death and places a significant burden on the health system due to the high costs associated with treating infected patients. In 2020, it was observed in studies that approximately 10 million patients had contracted TB, leading to around 1.5 million deaths [1,2]. Notably, an estimated 25% of the global population is believed to suffer from latent tuberculosis (LTBI), while extrapulmonary TB comprises about 20% of all TB cases, with intestinal tuberculosis making up 10% of these extrapulmonary cases [2,3,4].

Limited studies describe intestinal tuberculosis (ITB), which is defined as a secondary infection of the digestive tract by *Mtb*. This condition presents a severe prognosis that could lead to life-threatening complications. Studies highlight that some of the more severe complications include intestinal strictures that can lead to intestinal obstruction, as well as cases of perforation and intestinal bleeding [1,3,4,5]. There are some studies that outline the similarities between gastrointestinal tuberculosis and inflammatory bowel disease (IBD) due to the intestinal inflammation that can affect both the colonic and ileal segments. We note that, usually, in cases of ITB the appearance of the caecum and the ileocecal valve during colonoscopy often serves as a diagnostic indicator. However, it can be challenging to differentiate it from IBD, and, in such cases, biopsies are considered the gold standard to establish an accurate diagnosis. This review focuses mainly on ulcerative colitis (UC), Crohn’s disease (CD), and pouchitis, which are defined as chronic digestive disorders characterized by the inflammation of the intestinal mucosa and a tendency for relapses.

Studies indicate that the etiology of IBD is multifactorial, involving various influences, from environmental risk factors to microbial, genetic, and immune factors [6,7,8]. Current treatment guidelines for IBD focus on different strategies aimed at reducing inflammation and preventing additional extra digestive complications, such as arthritis and uveitis. However, practitioners often face treatment limitations, including significant relapse rates, drug resistance, and immune tolerance to biological therapy. Despite the use and study of new molecules, there is no perfect treatment to cure CD or UC [6,9].

As a result, researchers are actively exploring more effective treatment measures for IBD, and their recent studies have shown promising results in using FMT as a personalized therapy for recurrent or refractory IBD [6]. FMT represents the transfer of healthy fecal microbiota from a donor to a recipient, with the goal of improving microbial balance and modulating the immune response in the gut.

In this review, we want to emphasize that, due to some immunological similarities between these two diseases and the fact that they share common pathways, FMT may represent and important therapeutic approach to restore the native microbiome and improve the immunological status of patients. Therefore, this review aims to examine the potential application of FMT as an adjunctive therapy in intestinal tuberculosis, leveraging the extensive body of research supporting its effectiveness in IBD. By considering the shared features of dysbiosis and immune modulation in both diseases, we will explore the scientific rationale behind FMT as a complementary treatment approach for tuberculosis. In this paper, we also aim to emphasize that studies frequently explore the potential prior infection of *Mtb* in patients who later develop IBD.

## 2. Material and Methods

We realized a systematic review of the current studies following the PRISMA 2020 guidelines, applying specific inclusion and exclusion criteria to identify original reports and research related to the utilization of FMT as an adjunct therapy in TB. Our aim was to draw insights from cases and studies in IBD. The search was performed using prominent academic databases, including Google Scholar, Web of Science, PubMed, and Medscape, covering articles published until December 2022.

The inclusion criteria for the review encompassed peer-reviewed journals, studies, and meta-analyses addressing the involvement of FMT in TB, digestive tuberculosis, and IBD, as well as the implications of different immunological signaling pathways involved in inflammation and drug resistance. We excluded papers that were not subjected to peer review and studies related to other diseases, such as different types of cancer (for example colon cancer), graft-versus-host disease, etc.

The terms used for searching databases were “fecal”, “microbiota”, “microbiome”, “transplantation”, and “transferring” in association with other terms, such as “Tuberculosis”, “TB”, “Mtb”, “Intestinal Tuberculosis”, “ITB”, “Latent tuberculosis infection”, “LTBI”, and IBD descriptive terms, such as “Crohn disease”, “Crohn’s disease”, “inflammatory bowel disease”, “colitis”, “ulcerative colitis”, “IBD”, “CD”, or “UC”.

Additionally, terms related to both diseases, such as “immunomodulation,” “dysbiosis”, and “granuloma,” were included in the search to ensure a comprehensive exploration of the relevant literature (Figure 1).

## 3. Immunological Interplay: Unraveling the Connection between Tuberculosis and Crohn’s Disease

Crohn’s Disease and TB elicit robust immune responses mediated by TH1 cytokines, leading to the formation of granulomas. These responses involve the up-regulation of specific cytokines, like interferon-gamma (IFN-γ) and other interleukins (IL-12, and IL-23). Notably, these molecules are very important in containing and preventing the spread of *Mtb.* Studies outline the heightened vulnerability to the spread of atypical mycobacterial infections that present changes in signaling the IL-12/IL-23/IFN-γ pathway, which can result in severe infections and even multidrug resistance among infected patients [10,11].

Similarly, the diverse clinical manifestations seen in both CD and TB imply that variations in individual microbiota interactions might play a crucial role in the disease phenotype, and, also, an individual’s genetics could potentially influence the effectiveness of innate immune responses.

During the early stages of CD and TB, certain innate immune receptors, such as Nucleotide-binding oligomerization domain-2 (*NOD2*) and Toll-like receptors (TLRs), are believed to contribute to impaired innate immunity, leading to an abnormal response to antigens and triggers. These receptors are involved in recognizing specific components of a microbial origin, thereby initiating immune responses. Their involvement suggests a shared mechanism in the initial phases of both diseases [10,12,13]. Specific populations have shown susceptibility to CD due to the dis-regulation of the *NOD2 gene* [14]. Furthermore, *NOD2*, together with TLRs 2, 4, and 9, serves as a unique recognition system for detecting the presence of *Mtb*. Studies have emphasized the significant role of *NOD2* in the mononuclear cells of patients who develop CD and possess the homozygous 3020ins mutation in *NOD2*, a mutation also described in studies using *NOD2* knock-out mice as animal models [10,15].

Recent research has revealed that genetic variations in genes associated with the interleukin (IL)-23/IL-17 axis can impact the digestive mucosal integrity through their influence on the differentiation of Th17 cells, with implications in both diseases [16]. IL-17 and IL-22 play a crucial role in modulating the function of Th17 cells. IL-22 has an important role in maintaining mucosal immunity and a diversity in the bacterial flora, contributing to the biological barrier [17,18]. Additionally, IL-17 is involved in multiple immune responses, including the recruitment of neutrophils and the promotion of optimal inflammatory responses by Th1 cells [19,20]. Studies on animal models outline that IL-17 and IL-22 are also important with respect to the localization of T cells within lymphoid follicles in the lungs, facilitating the effective activation of macrophages and providing benefits in stimulating the immune response against *Mtb* [21]. Another important similarity between CD and ITB is the presence of granulomas, which can affect both the intestinal mucosa and peri-intestinal tissue, with IL-17’s clinical benefits playing a crucial role in limiting the formation of necrotic granulomas and thereby reducing the severity of TB disease [22,23,24,25]. Some researchers have proposed that enhancing the functions of mucosal-associated invariant T (MAIT) cells could represent one potential mechanism that influences the gut microbiota and brings its protective contribution to providing wide protection against *Mtb* colonization, including respiratory TB and extrapulmonary dissemination [26].

Other studies that describe the similarities between CD and ITB highlight the impact of gene polymorphisms within the IL-23/IL-17 axis on susceptibility to IBD and the development of ITB. In the same vein, genome-wide association studies conducted on Japanese and Korean populations have revealed associations between various genes that are signaling this pathway, which can predispose individuals to the development of IBD or, in the case of TB infection, to the development of ITB [16,17,18,19,20,21,22,23,24,25,26,27,28,29,30,31,32]. Previous studies involving population from China have noted that genetic polymorphism, specifically a single-nucleotide polymorphism in IL22 rs2227473, was found to be significantly correlated with susceptibility to developing TB [16,33]. Other studies that describe the susceptibility to developing IBD or ITB observed a correlation with genetic polymorphism in the IL6 promoter region, with rs1800795 being the only one of eleven screened polymorphic loci showing a correlation with ITB and possible implications for susceptibility to CD [34].

Furthermore, the IL1β promoter polymorphism rs1143627 has been found to be correlated to *Mtb* infection, with the genotype impacting the severity of TB and even developing ITB [16,35].

A case-control study, which included a total of 133 patients with intestinal tuberculosis (ITB), 128 patients with CD, and 500 healthy controls (HCs), was realized in order to determine the implication of specific single-nucleotide polymorphisms (SNPs) in different genes correlated to the IL-23/IL-17 axis and their clinical implications in signaling different pathways that influence the development of ITB and CD [16]. Current research is focused on exploring the implications of genetic polymorphisms in describing associations and susceptibility to both CD and TB, as well as the possible interconnections between the two diseases. Additionally, immunohistochemistry was realized to assess the expression of IL-22R1 in various causes of IBD and the potential connections between TB and CD [16].

The study also noted that the existence of the G allele in the IL22 gene promoter *SNP rs2227473* increases the risk of ITB, whereas an elevated expression of IL-22 serves as a protective factor against intestinal inflammation [16].

In patients with CD and TB, both serum and intestinal tissue samples exhibited significantly elevated levels of IL-22 expression, which also corresponded with disease activity [36]. Surprisingly, the investigation showed that IL-22R is up-regulated in epithelial cells and in Langhans giant cells; it also seems to be up-regulated in both TB and CD and plays a crucial role in granuloma formation. Notably, macrophage-derived Langhans giant cells, particularly those from individuals with intestinal tuberculosis, displayed high levels of IL-22R1 expression. This study record is crucial in understanding the impact that IL-22 presents in regulating adaptive immunity through its signaling of macrophages and stimulation of innate immunity [16]. Furthermore, IL-22 directly enhances macrophages, triggering the activation of TNF, which plays a crucial role in the immune response against TB [16,37]. TNF plays an important role in various chronic inflammatory disorders, also influencing CD [16,38]. However, commonly used anti-TNF antibodies, like Infliximab, which are beneficial for CD, can lead to unintended immunosuppression and reactivate *Mtb* in cases of LTBI [16,39,40].

In another study, individuals with a down-regulation of *IL-22* in bronchoalveolar lavage fluid (BALF) and *Mycobacterium avium complex* (*MAC*) infection displayed a predominant inflammatory response by neutrophils, and progressive radiological severity was also observed. Conversely, those with an up-regulation of *IL-22* in BALF presented a higher percentage of lymphocytes, which presented a protective role, resulting in a lower disease severity [41,42].

Recent research has outlined that when exposed to *Mtb* humans develop a distinct population of antigen-specific IL-22, which leads to the stimulation of the formation of CD4+ T cells characterized by a memory phenotype. Another relevant observation regarding the importance of *IL-22* is the fact that these cells were initially identified in blood samples that were stimulated with mycobacterial antigens from individuals exposed to *Mtb*. Studies note that patients with LTBI who experienced disease reactivation presented higher concentrations of *Mtb*-specific IL-22-producing CD4+ *cells* compared to patients with an active disease. This observation is consistent with the higher levels of IFNγ-producing Th1 cells observed during LTBI compared to active TB. Moreover, a specific genetic variation in the promoter of the IL-22 gene, which is associated with higher IL-22, Th1, Th22, and Th17 production in response to *Mtb* antigens, is more prevalent in control subjects compared to TB patients, suggesting a potential association with reduced susceptibility to TB [41,43]. We observed that in active TB fewer cases of ITB were encountered due to the stimulation of different immunological mechanisms. However, in cases of LTBI, more cases of ITB refractory to treatment were encountered, likely due to genetics, comorbidities, and other dysbiosis resulting from TB treatment. [41,43]

In cases where ITB is misdiagnosed as CD, the initiation of immunosuppressive therapy is a major contraindication because it can lead to TB dissemination and even result in life-threatening complications [44,45].

## 4. Dysbiosis Unveiled: Unraveling the Crucial Role and Comparative Analysis in Tuberculosis and Crohn’s Disease

The human gut microbiota is a highly intricate and diverse organ, hosting a diverse array of over 100 trillion commensal microorganisms. It is primarily characterized by the predominance of *Firmicutes* and *Bacteroidetes*, along with *Actinobacteria* and *Proteobacteria* in slightly lower proportions. Additionally, other significant phyla, such as *Verrucomicrobia*, *Fusobacteria*, and *Euryarchaeota,* contribute to the overall taxonomic composition [46,47].

Studies emphasize that the commensal microbiota plays an important role in regulating both adaptive and innate immunity through the production of small molecules known as metabolites. These metabolites can regulate the activation of immune system in response to pathogen stimulation, thereby modulating the host’s immune defense during disease [48]. The biological compounds that are released by the gut microbiome species have also been found to have direct microbicidal effects on pathogens by signaling the immune system [49].

Studies have shown that indole-3 propionic acide (IPA), produced by *Clostridium spores*, reduces the burden of *Mtb* in a mouse model and exhibits favorable pharmacokinetic properties [50]. The mechanism through which IPA influences *Mtb* is still under research, but studies suggest that it influences the production of tryptophan in *Mtb*, acting as a suppressor in disease dissemination [51].

Studies on animal models showed that the colonization of germ-free mice with *Bacteroides fragilis* induced the up-regulation of protective CD4+ T cells and restored the balance between Th1 and Th2 cytokines. Additionally, an increased production of IFN-γ and TNF-α was noted, leading to clinical benefits in the case of ITB [52]. Similarly, the instillation of *Clostridia* strains in mice resulted in an increased IL-10 secretion, leading to anti-inflammatory effects due to the stimulation of systemic and intestinal regulatory T cells (Tregs) [53].

In a human study investigating the interaction between inflammatory biomarkers and the gut microbiome in individuals with active TB and LTBI prior to anti-TB treatment, the authors noted that in active TB patients a decreased proportion of *Firmicutes/Bacteroidetes* and higher levels of *Bacteroidetes* in stool are associated with gut dysbiosis and trigger systemic pro-inflammation [54].

Regarding anti-TB therapy, studies have consistently demonstrated that it leads to a notable reduction in the population of healthy bacterial species within the gut microbiota of affected individuals, leading to an abnormal immunological response [55].

We observed that studies emphasize that anti-TB therapy results in persistent dysbiosis characterized by specific markers, including a decline in the *Clostridiales* population and other healthy bacteria, such as the *Firmicutes phylum*, *Clostridiales*, *Ruminococcus,* and *Faecalibacterium*, as well as a higher production of *Actinobacteria* and *Proteobacteria*. It is worth noting that within the *Proteobacteria* category there are microbes like *Escherichia*, *Salmonella, Yersinia,* and *Helicobacter*, which can enhance intestinal inflammation. This kind of dysbiosis has also been described in CD patients [56,57]. The current literature emphasizes the importance of healthy bacterial colonization, which leads to improvements in the immune system. We also observed studies on patients with melanoma and other types of cancers that show the crucial role of FMT in restoring chemo-sensitivity and improving overall survival. A study on an animal model outlined that gut dysbiosis, characterized by a high prevalence of *Bacteroides* species and a lower prevalence of *Firmicutes,* has been associated with an elevation in IL-10 levels and a diminished response to *Mtb* vaccination [58].

Notably, recent studies indicate that dysbiosis can persist for an extended period of time, even after discontinuing treatment. Therefore, alterations in the taxonomic composition and the reduced microbial richness in the gut microbiota can persist for up to 1–3 years (chronic effects) [59] and, in cases of multidrug-resistant TB, for up to 3–8 years after recovery and the cessation of treatment [60].

This imbalance of gut microbes disrupts microbiota metabolic functions and is linked to an increased vulnerability to immune-related conditions, including IBD and allergies, which have seen a notable increase over the past few decades [61,62,63].

A cohort study that recruited 22 patients, comprising 6 individuals with ITB and 16 individuals witch active CD, aimed to investigate the changes in the gut microbiota in patients with ITB and compare the microbial composition between ITB and CD, highlighting the similarities between these conditions. Both are characterized by chronic intestinal inflammation that affects the digestive tract and are prone to abnormalities in mucosal immune responses [64].

Another study that included 71 gut analyses from individuals diagnosed with ITB and CD and healthy controls (HCs) conducted *16S rRNA gene* sequencing in order to analyze the gut microbiota [64]. The results indicate that the most prevalent phyla in both groups were *Firmicutes*, *Bacteroidetes*, and *Proteobacteria*, but their distribution varied between ITB and HC. In HC samples, *Firmicutes* had a higher prevalence, whereas *Proteobacteria* dominated in ITB patients, and healthy bacteria like *Firmicutes* showed decreased levels [64].

Subsequently, the variations in microbial composition between active CD and HC at three taxonomic levels were also examined. Intriguingly, individuals with CD exhibited higher levels of *Proteobacteria* and lower levels of *Firmicutes*, which were quite similar to the alterations observed in ITB compared to HC [64].

The abundance of bacteria responsible for producing short-chain fatty acids (SCFAs), such as *Faecalibacterium*, *Roseburia*, and *Ruminococcus,* was decreased in ITB patients compared to the HC group. Conversely, the presence of *Klebsiella* and *Pseudomonas* was found to be enriched in ITB patients. A decrease in the abundance of several SCFA producers, such as *Roseburia* and *Ruminococcus*, was also noted in CD compared to the HCs [64].

Comparing the taxonomic profiles between ITB and CD groups, distinct changes in the abundance of various taxa were observed. At the phylum level, ITB samples had a lower relative abundance of *Firmicutes* compared to CD. Furthermore, at the family level, both *Ruminococcaceae* (belonging to the Firmicutes phylum) and *Bacteroidaceae* were significantly reduced in ITB compared to CD [64] (Table 1).

## 5. Harnessing the Power of Fecal Microbiota Transplantation (FMT) in Inflammatory Bowel Disease (IBD): A Promising Therapeutic Avenue

Traditional treatment approaches for IBD have predominantly centered around inflammation reduction. Despite ongoing development and updates to these treatment regimens, there are still limitations, including high relapse rates, immune tolerance, and drug resistance [6,9].

One widely accepted observation is that patients with IBD have altered gut microbiota [6].

Therefore, there has been a growing interest among researchers in exploring therapeutic alternatives to improve and restore the homeostasis of the gut microbiota in recent years.

One such approach that has gained attention among IBD researchers is FMT, which has already been successfully used for managing recurrent *Clostridioides difficile* infection (CDI), with strong recommendations in the treatment guidelines of the United States and Europe for CDI cases [65,66].

FMT represents an innovative therapeutic approach that aims to correct dysbiosis by restoring a healthy, diverse microbiota from healthy individuals to the patient, thus restoring a functional gut ecosystem, including viruses and fungi with bacteriophage proprieties, in order to maintain a healthy microbiota [67]. It has also been explored in various disease fields [6], including enhancing chemotherapy sensitivity, especially in patients with advanced melanoma receiving anti-PD-1 immunotherapy [6,68,69].

As a result, there has been an increase in clinical studies investigating the efficacy of FMT in the treatment of refractory IBD.

So far, many studies have been conducted to evaluate the effectiveness of FMT in inducing remission in UC (Table 2) [6,70,71,72,73,74,75,76,77,78,79,80].

Moayyedi et al. conducted a study involving 75 patients with mild to severe UC. The study group received FMT via enemas from donors, while the control group received a placebo treatment. The study outlined that the patients receiving FMT achieved clinical remission compared to the control group, with statistically significant results (*p* = 0.03) [6,77].

Another study realized by Paramsothy et al. included 81 patients with mild to moderate UC, with 41 patients included in the study group and 40 in the control group. The results revealed a significantly higher rate of endoscopic remission in the study group compared to the control group at week 8 (*p* = 0.021) [6,78].

Costello et al. also reported a significantly better treatment effect in the study group, which included 38 patients with moderate UC who received FMT, compared to the control group with 35 patients in the placebo group. After a two-month follow-up, 12 patients (32%) in the FMT group achieved clinical and endoscopic remission, while only 3 out of 35 patients achieved complete remission (*p* = 0.03) [6,79].

These studies mentioned above indicate that FMT has shown effectiveness in inducing remission in IBD. However, further research is needed to establish guidelines for the application of FMT in IBD. Clinical trials have demonstrated higher rates of clinical and endoscopic remission in patients who received FMT compared to those who received a placebo. These findings suggest that FMT could be a promising therapeutic approach for managing UC.

Several studies have also investigated the effectiveness of FMT in CD patients, demonstrating positive outcomes, such as higher rates of steroid-free remission and improvements in clinical targets [70]. In a recent pilot randomized controlled trial (RCT), the impact of FMT as a maintenance treatment for CD was evaluated. The study included 18 CD patients who received FMT results and showed a higher rate of clinical remission in the FMT group (57.1%) compared to the placebo group (33.3%) at the 24-week mark [70].

Furthermore, the trial also recorded improvements in the CD Endoscopic Index of Severity in the FMT group at 6 weeks, while no significant improvement was seen in the control group [70].

Notably, the study identified that the two patients in the FMT group who experienced early relapse were the ones who did not exhibit an engraftment of the donor microbiota by week 6. These promising findings suggest that FMT may be an effective approach for maintaining remission in CD patients. However, further research, such as a Phase III RCT, is needed to validate these results and assess the long-term effectiveness and safety of FMT in CD treatment [70].

In an RCT involving 21 patients, single-dose FMT was compared to a placebo in individuals who had achieved remission with steroids. The FMT group exhibited a higher rate of steroid-free clinical remission compared to the control group (*p* = 0.23) [71].

In another RCT involving 31 patients, a two-dose FMT regimen was evaluated. The study found that the rate of clinical remission after two months was 36%. Furthermore, the study noted that there was no significant difference in the rate of endoscopic remission when FMT was administered via gastroscopy or colonoscopy. This finding is important for clinicians to consider when choosing the FMT delivery method. It is worth highlighting that multiple FMT treatments resulted in higher clinical and endoscopic remission rates compared to single FMT, and remission was achieved sooner with multiple FMT instillations. [71].

Furthermore, current research has outlined similar success rates for various types of microbiota transplantation. These include washed microbiota transplantation, spore transplantation, which includes the transplantation of bacteriophages (likely in numbers similar to bacteria), as well as fungi and other microbes that can collectively contribute to the restoration of a healthy microbiota. According to Koch’s postulate, which describes the involvement of a single pathogen in dysbiosis leading to both IBD and ITB, restoring a microbiota rich in beneficial microbes, virions, and fungi effectively reinstates a healthy gut microbiota. This, in turn, has the potential to regulate the innate immunological response in cases of IBD and ITB [6,55,71]. New perspectives are emerging, focusing on fecal virome transplantation (FVT), which involves the use of bacteria-filtered stool, obtained through centrifugation. This approach aims to enhance safety and reduce the risk of adverse effects, especially in immunosuppressed patients, such as those with ITB or individuals undergoing anti-TNF therapy for IBD. FVT serves as a promising alternative to mitigate the risk of bacteremia and sepsis development in these vulnerable patient populations [6,51,55,71].

Another study aimed to assess the efficacy of FMT in achieving clinical targets in patients with CD. A total of 174 patients completed the follow-up, and the results demonstrated significant improvements across various clinical parameters post-FMT. Specifically, the study observed a reduction in abdominal pain, fewer cases of infection and fever, a decrease in diarrhea, and an improved drainage of fistulas. At 1 month after FMT, a substantial percentage of patients experienced relief, with 72.7% (101/139) reporting reduced abdominal pain, 61.6% (90/146) experiencing improved diarrhea, 76% (19/25) noting relief from hematochezia, and 70.6% (12/17) reporting decreased fever [72].

To sum up, all these studies highlighted the potential of FMT as an effective therapeutic alternative for managing both CD and UC. In CD, FMT has shown a higher rate of steroid-free clinical remission and improvements in the CD Endoscopic Index of Severity. Similarly, in UC, FMT has demonstrated higher rates of clinical and endoscopic remission compared to placebo groups. These findings suggest that FMT holds promise for inducing remission and improving clinical targets in patients with IBD. However, further research, including larger-scale trials and long-term assessments, is necessary to validate these results and optimize the protocols for FMT in clinical practice.

## 6. Unveiling the Potential of Fecal Microbiota Transplantation in Tuberculosis Treatment: Insights from Mouse Studies and the Path to Clinical Applications

Recent studies on mice infected with *Mtb* have unveiled intriguing insights into the significance of the gut microbiota and the potential of FMT as a treatment for ITB.

In a recent study that investigated the role of the gut microbiota in regulating the pathogenesis of ITB, researchers disrupted the gut microbiota in *Mtb*-infected mice using antibiotics, resulting in notable alterations in the gut microbiota composition. Interestingly, these changes were associated with ITB and the dissemination of the pathogen to the spleen and liver. To further investigate the impact of these antibiotic-induced alterations, FMT was performed, which restored the gut microbiota, reduced the complications of ITB, and prevented pathogen dissemination [18].

Another study provided further evidence of the detrimental effects of antibiotics on the healthy gut microbiota in *Mtb*-infected animals. This disruption compromised mouse immunity, leading to increased pathogen dissemination to other organs and causing ITB. These changes in the gut microbiota influenced the immune responses to ITB, leading to the dis-regulation of Tregs expansion and a decrease in the frequencies of IFN-γ- and TNF-α-producing Th1 cells. Remarkably, FMT restored TB immunity and prevented the dissemination of ITB to the spleen and liver [73,74,75,76].

Studies conducted on mice infected with ITB have provided valuable insights into the impact of gut microbiota disruption on TB infection outcomes and the potential of FMT as a therapeutic intervention. These findings highlight the significance of the gut microbiota in TB pathogenesis and the potential of manipulating the microbiota through FMT to enhance ITB treatment outcomes, given its potential in reversing dysbiosis caused by TB antibiotics [73,74,75,76].

However, in spite of the valuable insights gained from recent studies conducted on mice infected with *Mtb* and the potential of FMT in TB treatment, it is important to acknowledge the limitations of the current research, which primarily relies on murine models. Further studies involving human subjects are crucial to validate and translate these findings into clinical applications. Understanding the impact of FMT on the human gut microbiota and its potential as a therapeutic approach in TB requires additional investigation and clinical trials. Such studies will provide more comprehensive evidence and pave the way for the development of targeted and personalized FMT-based interventions in the management of TB [73,74,75,76].

## 7. Conclusions

In conclusion, this review has examined the potential benefits of FMT as an adjunct therapy in TB, drawing from its efficacy in IBD. FMT has shown promising results in restoring the gut microbial balance and modulating immune responses in IBD patients. Given the similarities in immunomodulation and dysbiosis between IBD and TB, this review hypothesized that FMT could also provide therapeutic benefits as an adjunct treatment in TB. By analyzing the existing literature on FMT in both IBD and TB, shared mechanisms and potential implications for FMT in TB management were highlighted. The findings contribute to the understanding of FMT’s potential role in TB treatment and provide insights for future research and clinical applications.

Indeed, recognizing the potential therapeutic benefits of FMT in both IBD and TB offers hope for patients facing refractory disease or developing resistance to conventional medical treatments. Restoring a healthy gut microbiome through FMT may help reduce inflammation, mitigate fistulas, and decrease the need for surgeries in patients with CD, while also offering new possibilities for managing TB.

However, further studies are essential to elucidate the specific mechanisms through which FMT influences the pathogenesis of ITB and to establish optimal protocols for its application in ITB patients. Given the limited research on FMT in active/inactive TB and ITB, additional animal studies should be conducted to assess safety, doses, frequency, administration routes, donor selection, and efficacy. Despite these challenges and the scarcity of studies on FMT in ITB, the concept of using FMT as an adjunct therapy in ITB is promising and should be explored further. We emphasize that, although there are very few studies on FMT in ITB and further animal studies should be carried out, the similarities in the immunological mechanisms and the clinical positive outcomes of FMT in IBD highlight new research directions for FMT in ITB.

The integration of FMT into the comprehensive management of ITB could potentially enhance treatment outcomes, reduce drug resistance, and mitigate the side effects of conventional therapies. Future research endeavors should focus on well-designed clinical trials to assess the safety, efficacy, and long-term benefits of FMT in TB patients. Ultimately, the findings from such studies could pave the way for the development of novel treatment strategies that harness the therapeutic potential of the gut microbiota in combating ITB, improving patient outcomes, optimizing drug management, and preventing multidrug resistance.

## Figures and Tables

**Figure 1 pathogens-12-01149-f001:**
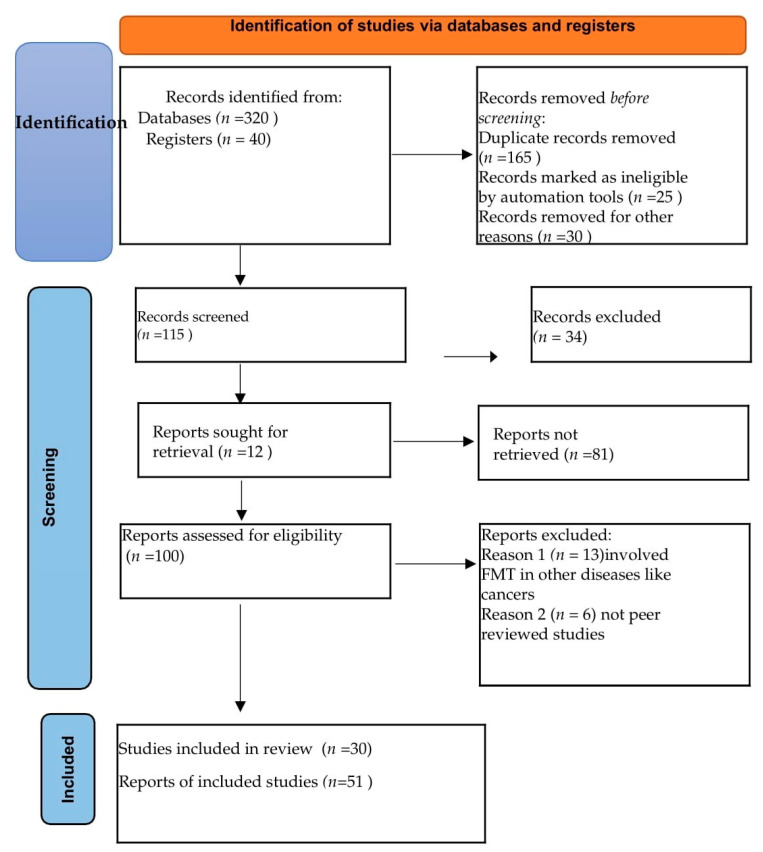
PRISMA flow diagram.

**Table 1 pathogens-12-01149-t001:** Comparison of intestinal microbiota composition in ITB (intestinal tuberculosis) group, CD (Crohn’s disease) group, and HC (healthy controls) group and similarities between ITB and CD.

	HC	ITB	CD
*Firmicutes*	High bacterial level	Low bacterial level	Low bacterial level
*Proteobacteria*	Low bacterial level	High bacterial level	High bacterial level
*Enterobacteriaceae*	Low bacterial level	High bacterial level	High bacterial level
*Lachnospiraceae*	High bacterial level	Low bacterial level	Low bacterial level
*Ruminococcus*	High bacterial level	Low bacterial level	Low bacterial level
*Roseburia*	High bacterial level	Low bacterial level	Low bacterial level

**Table 2 pathogens-12-01149-t002:** Summary of clinical trials on fecal microbiota transplantation (FMT) in ulcerative colitis (UC).

	Moayyedi et al. [77]	Paramsothy et al. [78]	Costello et al. [79]	Haifer et al. [80]
Number of patients	75 (FMT group: 38, placebo group: 37)	81 (FMT group 41/placebo group 40)	73 (FMT group 38/placebo group 35)	35 (FMT group 15/placebo group 20)
Primary endpoint (FMT vs. placebo)	CR (clinical response) + ER (endoscopic response) at week 7, 24 vs. 5%, *p* = 0.03	CR (clinical response) + ER (endoscopic response) at week 8, 27 vs. 8%, *p* = 0.02	CR (clinical response) + ER at week 8, 32 vs. 9%, *p* = 0.03	CR (clinical response) + ER (endoscopic response) at week 8, 53 vs. 15%, *p* = 0.027
General clinical remission (FMT group vs. placebo group)	24 vs. 5%, *p* = 0.03	44 vs. 20%, *p* = 0.02	47 vs. 17%, *p* = 0.01	73 vs. 25%, *p* = 0.0045

## Data Availability

Not applicable.

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
