# Peer review of "Exploring the Potential of Fecal Microbiota Transplantation as a Therapy in Tuberculosis and Inflammatory Bowel Disease"

_pathogens, 2023, doi:10.3390/pathogens12091149_

Round 1
Reviewer 1 Report
This manuscript present a review of therapy-supporting potencial of FMT in IBD and tuberculosis. The work deals with an important topic concerning the improvement of the effectiveness of tuberculosis treatment.
The manuscript generally looks like it's still a working draft. Requires text editing, correctly used abbreviations, reference list, proper table captions. Without necessary corrections of the text the manuscript should not be published in this form.
Generally:
- the authors repeat the use of the abbreviation each time in brackets (for example TB, FMT, CD, ITB, SNP) in the text instead of using the abbreviation;
- all human gene names should be written in italics (for example: page 3 line 103 "NOD2 gene");
- names of bacteria should be written in italics (for example: page 6, lines 235-236, line 238, page 7, lines 269-270);
- What present Table 1. actually? Some selected different taxonomic levels are only showed but description of the Table is "... composition" - it is not fully composition, but only some differences. Furthermore, the abbreviations used in the name of the table should be explained;
- the list of references does not have a consistent form
Generally the english language is good, but some fragments need to be improved, sa page 4 lines 124-131 the word "susceptibility" is used too often.
Author Response
We appreciate the thoroughness of the distinguished reviewer’s evaluation. We have proofread the manuscript, and have corrected all formatting errors throughout it, we also improved table 1 and explained the abbreviations in the tables.
- The list of references does not have a consistent form
We thank the distinguished reviewer for raising this issue, we revised the references.
Reviewer 2 Report
Brief summary:
In this literature review by Boicean et al., they go through the common denominators between IBD and TB to suggest that FMT should also be investigated as a tool to treatment humans suffering from TB infections.
Broad comments and major concerns:
I surely find it relevant to seek FMT as a tool to restore the GM and thereby indirectly treating TB, however, I remain to have some concerns that needs to be addressed.
Major concerns
1. Reference [71] is not listed in the reference list, and since this reference is used quite a lot, I cannot evaluate the paragraphs involving the reference before its listed correctly.
2. Its for me not possible to really understand which kind of TB that the authors would like to treat with FMT. In the introduction its mentioned that 10% of TB-cases is intestinal TB (ITB). Here FMT seems to make quite good sense, but would the authors also treat the remaining 90% of TB cases that constitute other types? Please, specify exactly what you want. And if other than ITB should be treated with FMT, please suggest a hypothesis of the mechanisms that can explain how FMT should treat e.g. infections in lung tissues?
3. The clinical and immunological perspective of the review appears for me really good, but, it would be great if the authors could include a microbiologist as an additional co-author to improve the microbiology perspective. E.g.
a. Its not mentioned at all that the transfer of FMT includes viruses, including bacteriophages that are likely as many in numbers as bacteria, but also fungi, archaea and other microbes. Thus the many phages transferred will likely also play a role in the efficacy – e.g. showed by Ott et al. 2017 that treated C. difficile patient with bacteria-free filtrates (Ott et al., 2017) aka Fecal Virome Transplantation (FVT). That represents safer alternative to FMT (Rasmussen et al., 2020) through the removal of the bacterial component.
For that reason FVT should also be mentioned as a tool for treating e.g. immune suppressive patients, as mentioned in the conclusion is mentioned as a problem.
b. Its not mentioned whether there are strain-difference in the active vs non-active cases of TB? Is that due to strains being more pathogenic than others? Or just human genetics, age, health status? Or all combined?’
c. And as a minor part, many places the bacterial names are not in Italic. The newest recommendation from IJSEM is pretty easy – just make all names italic.
4. The reference [6] at line 46 seems wrong. This paper is about COVID-19 and C. difficile?. And other studies also shows that FMT is not necessarily so efficient in treating IBD (Federici et al., 2022; Tran et al., 2018).
5. In case FMT was used against ITB, what would be the hypothesis that could explain that FMT would work? Re-wilding with “good” bacteria”? This hypothesis is lacking.
6. Fine to suggest clinical studies based on the few TB-FMT studies, but it needs to be clear whether the authors mean ITB or TB in general? And before going to humans, maybe additional animal studies should be performed in term of evaluating safety, dose, frequency, administration route, donor, and efficacy. So a clinical trial may not be the first next step.
Minor concerns:
· Line 66: “cancer” appears twice.
· Line 425: They reversed the dysbiosis, but how successful? And how was that defined? The reference in the end of the paragraph is missing.
· Line 454: FMT is by principle not modern (used by ancient Chinese medical doctors).
· Figure 1: low resolution of the figure.
· Some abbreviations are explained more than once. Only do it once.
Federici, S., Kredo-Russo, S., Valdés-Mas, R., Kviatcovsky, D., Weinstock, E., Matiuhin, Y., Silberberg, Y., Atarashi, K., Furuichi, M., Oka, A., Liu, B., Fibelman, M., Weiner, I. N., Khabra, E., Cullin, N., Ben-Yishai, N., Inbar, D., Ben-David, H., Nicenboim, J., … Elinav, E. (2022). Targeted suppression of human IBD-associated gut microbiota commensals by phage consortia for treatment of intestinal inflammation. Cell, 2879–2898. https://doi.org/10.1016/j.cell.2022.07.003
Ott, S. J., Waetzig, G. H., Rehman, A., Moltzau-Anderson, J., Bharti, R., Grasis, J. A., Cassidy, L., Tholey, A., Fickenscher, H., Seegert, D., Rosenstiel, P., & Schreiber, S. (2017). Efficacy of sterile fecal filtrate transfer for treating patients with Clostridium difficile infection. Gastroenterology, 152(4), 799-811.e7. https://doi.org/10.1053/j.gastro.2016.11.010
Rasmussen, T. S., Koefoed, A. K., Jakobsen, R. R., Deng, L., Castro-Mejía, J. L., Brunse, A., Neve, H., Vogensen, F. K., & Nielsen, D. S. (2020). Bacteriophage-mediated manipulation of the gut microbiome - promises and presents limitations. FEMS Microbiology Reviews, 44(4), 507–521. https://doi.org/10.1093/femsre/fuaa020
Tran, V., Phan, J., Nulsen, B., Huang, L., Kaneshiro, M., Weiss, G., Ho, W., Sack, J., Ha, C., Uslan, D., & Sauk, J. S. (2018). Severe Ileocolonic Crohnʼs Disease Flare Associated with Fecal Microbiota Transplantation Requiring Diverting Ileostomy. ACG Case Reports Journal, 5(12), e971-4. https://doi.org/10.14309/crj.2018.97
Nothing to add - this should be the responsibility of the journal to improve grammar.
Author Response
- Reference [71] is not listed in the reference list, and since this reference is used quite a lot, I cannot evaluate the paragraphs involving the reference before its listed correctly.
We thank the distinguished reviewer for raising this important issue , we included the reference list.
- Its for me not possible to really understand which kind of TB that the authors would like to treat with FMT. In the introduction its mentioned that 10% of TB-cases is intestinal TB (ITB). Here FMT seems to make quite good sense, but would the authors also treat the remaining 90% of TB cases that constitute other types? Please, specify exactly what you want. And if other than ITB should be treated with FMT, please suggest a hypothesis of the mechanisms that can explain how FMT should treat e.g. infections in lung tissues?
We thank the distinguished reviewer for raising this important question, we found immunological similarities in studies on murines, as well as on stool metagenomic sequencing between ITB and CD, we highlighted this idea, thank you very much for your suggestion. We also insisted on the hypothesis of the mechanisms that can explain how FMT could improve ITB dysbiosis in case of multidrug resistance by repopulating the microbiota and achieve a diversity in healthy bacteria, bacteriophages, fungi and viruses.
- The clinical and immunological perspective of the review appears for me really good, but, it would be great if the authors could include a microbiologist as an additional co-author to improve the microbiology perspective. E.g.
- Its not mentioned at all that the transfer of FMT includes viruses, including bacteriophages that are likely as many in numbers as bacteria, but also fungi, archaea and other microbes. Thus the many phages transferred will likely also play a role in the efficacy – e.g. showed by Ott et al. 2017 that treated C. difficile patient with bacteria-free filtrates (Ott et al., 2017) aka Fecal Virome Transplantation (FVT). That represents safer alternative to FMT (Rasmussen et al., 2020) through the removal of the bacterial component.
For that reason FVT should also be mentioned as a tool for treating e.g. immune suppressive patients, as mentioned in the c
We appreciate the thoroughness of the distinguished reviewer’s evaluation and suggestions in order to improve our manuscript, we included the suggested authors and ideas.
- Its not mentioned whether there are strain-difference in the active vs non-active cases of TB? Is that due to strains being more pathogenic than others? Or just human genetics, age, health status? Or all combined?’
We thank the distinguished reviewer for raising this issue in need of clarification, we clarified this perspective in the manuscript.
- And as a minor part, many places the bacterial names are not in Italic. The newest recommendation from IJSEM is pretty easy – just make all names italic.
We thank you very much for this suggestion, we revised it through the manuscript.
- The reference [6] at line 46 seems wrong. This paper is about COVID-19 and C. difficile?. And other studies also shows that FMT is not necessarily so efficient in treating IBD (Federici et al., 2022; Tran et al., 2018).
- We thank you very much for for raising this important question, we revised the references.
- In case FMT was used against ITB, what would be the hypothesis that could explain that FMT would work? Re-wilding with “good” bacteria”? This hypothesis is lacking.
- We appreciate this suggestion, we highlighted in the manuscript the hypothesis of enriching the microbiota, re-wilding it with good bacteria and enssuring an healthy diversity of the gut microbiome.
- Fine to suggest clinical studies based on the few TB-FMT studies, but it needs to be clear whether the authors mean ITB or TB in general? And before going to humans, maybe additional animal studies should be performed in term of evaluating safety, dose, frequency, administration route, donor, and efficacy. So a clinical trial may not be the first next step.
We appreciate the thoroughness of the distinguished reviewer’s evaluation, we outlined that further studies on animal models should be carried out in order to asses guidelines to use it in clinical practice, however we consider this idea could open new perspectives of research.
Minor concerns:
- Line 66: “cancer” appears twice.
- Line 425: They reversed the dysbiosis, but how successful? And how was that defined? The reference in the end of the paragraph is missing.
- Line 454: FMT is by principle not modern (used by ancient Chinese medical doctors).
- Figure 1: low resolution of the figure.
- Some abbreviations are explained more than once. Only do it once.
We appreciate the thoroughness of the distinguished reviewer’s evaluation. We have proofread the manuscript, and have corrected all formatting errors throughout it and also included the suggested authors and ideas, we thank you once again
Round 2
Reviewer 2 Report
I thanks the authors for adressing my concerns properly. I have no further comments.
Good luck with your future research.
As mentioned, only minor editing would be necessary.